# Epiploic Adipose Tissue (EPAT) in Obese Individuals Promotes Colonic Tumorigenesis: A Novel Model for EPAT-Dependent Colorectal Cancer Progression

**DOI:** 10.3390/cancers15030977

**Published:** 2023-02-03

**Authors:** Rida Iftikhar, Patricia Snarski, Angelle N. King, Jenisha Ghimire, Emmanuelle Ruiz, Frank Lau, Suzana D. Savkovic

**Affiliations:** 1Department of Pathology and Laboratory Medicine, Tulane University School of Medicine, New Orleans, LA 70112, USA; 2Department of Surgery, Louisiana State University Health Sciences Center, New Orleans, LA 70112, USA

**Keywords:** obesity, epiploic adipose tissue (EPAT), colon cancer

## Abstract

**Simple Summary:**

The role of epiploic adipose tissue (EPAT), understudied fat appendages attached to the colon, in obesity-facilitated colorectal cancer (CRC) is unexamined. In our novel microphysiological system, EPAT obtained from obese individuals, unlike EPAT from lean, attracts colon cancer cells’ intrusion and enhances their migration and growth. Conditioned media from this model mediated gene expression in colon cancer cells that are linked to metabolic and tumorigenic remodeling. This EPAT-mediated transcriptional signature defines transcriptomes of human colon cancer. These findings highlight a tumor-promoting role of EPAT, a metabolic tissue, in the colon of obese individuals and establishes a platform for exploration of involved mechanisms and development of effective treatments.

**Abstract:**

The obesity epidemic is associated with increased colorectal cancer (CRC) risk and progression, the mechanisms of which remain unclear. In obese individuals, hypertrophic epiploic adipose tissue (EPAT), attached to the colon, has unique characteristics compared to other fats. We hypothesized that this understudied fat could serve as a tumor-promoting tissue and developed a novel microphysiological system (MPS) for human EPAT-dependent colorectal cancer (CRC-MPS). In CRC-MPS, obese EPAT, unlike lean EPAT, considerably attracted colon cancer HT29-GFP cells and enhanced their growth. Conditioned media (CM) from the obese CRC-MPS significantly increased the growth and migration of HT29 and HCT116 cells (*p* < 0.001). In HT29 cells, CM stimulated differential gene expression (hOEC_867_) linked to cancer, tumor morphology, and metabolism similar to those in the colon of high-fat-diet obese mice. The hOEC_867_ signature represented pathways found in human colon cancer. In unsupervised clustering, hOEC_867_ separated transcriptomes of colon cancer samples from normal with high significance (PCA, *p =* 9.6 × 10^−11^). These genes, validated in CM-treated HT29 cells (*p* < 0.05), regulate the cell cycle, cancer stem cells, methylation, and metastasis, and are similarly altered in human colon cancer (TCGA). These findings highlight a tumor-promoting role of EPAT in CRC facilitated with obesity and establishes a platform to explore critical mechanisms and develop effective treatments.

## 1. Introduction

The obesity epidemic affects half a billion individuals worldwide [1]. Obesity is directly associated with increased colorectal cancer (CRC) risk, progression, recurrence, resistance to therapy, and mortality [2,3]. CRC, the second leading cause of cancer-related deaths worldwide, is initiated and driven by complex intracellular and extracellular remodeling [4,5]. We have demonstrated that colonic tumorigenesis augmented by obesity is mediated by increased lipid metabolism in colonic cells and surroundings [6,7,8]. Understanding these obesity-driven tumorigenic processes will promote further exploration of critical mechanisms and the development of more effective treatments. 

Emerging findings revealed that epiploic adipose tissue (EPAT), visceral fat appendages attached to the colon, is linked to obesity-mediated systemic pathobiology [9]. Adult individuals have about 50–100 of these EPAT appendages, which are ~1.5 cm thick and 3.5 cm long [10]. EPAT provides blood supply for the colon, supports colonic absorption, and delivers nutrition to the colon during starvation [11,12]. It has been speculated that EPAT is involved in host–microbe interactions and may be regulated by paracrine factors [13]. In obese individuals, EPAT is enlarged and is linked to colonic pathobiology such as diverticulitis and epiploic appendagitis [14,15]. Recently, Krieg et al. assessed abdominal fat in a large number of obese individuals undergoing gastric bypass surgery and demonstrated unique characteristics of EPAT compared to other adipose tissue [9]. Given these lines of evidence, we hypothesized that this understudied fat serves as a tumor-promoting tissue in the colon of obese individuals. In order to study the role of EPAT in colorectal cancer progression, it was necessary to establish a novel model since neither in vitro nor in vivo models specific to this fat are available. A recently developed microphysiological system (MPS) appeared to be suitable for studying the role of adipose tissues in obesity processes [16,17,18]; thereby, we utilized this platform to establish a novel MPS specific for EPAT-dependent colorectal cancer.

## 2. Materials and Methods

### 2.1. Human Samples

Human epiploic adipose tissue (EPAT) samples were obtained from local patients undergoing surgery unrelated to colonic inflammation and cancer from the larger New Orleans metropolitan area. Patients (ages 16–67) were a mix of female and male, African American and Caucasian, with Body Mass Index (BMI), as calculated (kilograms/meters^2^), ranging from 17.6 to 40.7. The de-identified patient samples used in this study were approved by the institutional review board (IRB, protocol number 867) at Tulane University, which waived the requirement for informed consent for sample collection. 

Publicly available transcriptomes from human colon cancer patients comprised control (*n =* 23) and colonic tumor samples (*n =* 198) (GSE4183; GSE141174). Additionally, publicly available TCGA transcriptomic data was obtained from colon cancer patients (normal colon (*n =* 41) and tumor (*n =* 457)). These data were acquired using NCBI’s GEO2R.

### 2.2. Mouse Model for High-Fat Diet Obesity and Colonic Tumorigenesis (Transcriptomic Data)

C57BL/6J mice (6 weeks old) were housed at Tulane University School of Medicine according to the guidelines of the Tulane Institutional Animal Care and Use Committee (protocol number 1161). One group of mice was maintained on a standard chow diet (RD), and the other on a high-fat chow diet (HFD) (60% kcal/fat) (D12492, Research Diets, New Brunswick, NJ). In addition, colonic tumors were induced in experimental mice by a single azoxymethane (AOM, Sigma, St. Louis, MO, USA) intraperitoneal injection of 10 mg/kg, followed by three separate 5-day cycles of 2.5% dextran sulfate sodium (DSS, MP Biomedicals, San Diego, CA, USA) added to drinking water, as we described before [8]. Transcriptomic analysis was performed after RNA-seq from the colons of these mice (*n =* 3 for each group) and is available through NCBI’s Sequence Read Archive (SRP093363) [8]. 

### 2.3. Cells

Human colon cancer cells HT29 (ATCC, Manassas, VA, USA), GFP-tagged HT29 (Genecopoeia, Rockville, MD, USA), and HCT116 (ATCC, Manassas, VA) were propagated in complete McCoy’s 5A media (Sigma, St. Louis, MO) containing 10% fetal bovine serum (FBS) (Peak Serum, Wellington, CO, USA). EPAT-derived stromal cells (ESC) from human EPAT were isolated by collagenase digestion of tissue, vigorous washing, and selection via cell adherence [19,20]. ESC were propagated using DMEM containing 10% fetal bovine serum and 10% penicillin/streptomycin (Gibco, Waltham, MA, USA). Cancer cells were serum-starved overnight prior to experimental procedures and were serum-starved for three days to synchronize cells in the cell cycle prior to treatments. 

### 2.4. Epiploic Colonic Microphysiological System (CRC-MPS) and Conditioned Media 

EPAT was obtained from obese or lean patients undergoing abdominal surgery unrelated to colonic diseases (inflammation or cancer). From EPAT, isolated adipose stromal cells (ESC) were used to sandwich CRC-MPS (containing EPAT-isolated adipocytes and human colonic cells) (Figure 1A). Fresh EPAT was physically minced, and 200 µL of their cell suspension was used for each CRC-MPS well. HT29-GFP cells, cultured independently, were trypsinized, and 200,000 cells were added to each CRC-MPS well. Two ESC sheets are needed for each CRC-MPS well. On the first ESC sheet, grown on a 6-well plate, EPAT and HT29-GFP cells were added. A second ESC monolayer was grown on a thermosensitive polymer (poly(N-isopropylacrylamide), Nunc UpWell 6-well dish, 174902). 3D-printed plungers loaded with a hydrogel (12% Gelatin B, ~225 bloom, G9382, Sigma, St. Louis, MO) were gently pressed onto the ESC monolayer, and incubated at room temperature, then at 4 °C, allowing the ESC layer to be lifted off the plate. The plunger carrying these ESC was then placed on top of EPAT and HT29-GFP cells that were added to the first ESC monolayer to create a sandwich (Figure 1A). Colon cancer cell status in CRC-MPS was assessed daily under the microscope, and GFP signal from acquired images was quantified by pixel area per field of view. These CRC-MPS, kept in DMEM and 10% FBS at 37 °C, were maintained for 5 days. Their 24 h growth media, diluted 1:5 with serum-free McCoy’s 5A media, was used as a conditioned media (CM) to treat colon cancer cells. Two cell lines were used to increase rigor and reproducibility. 

### 2.5. BrdU and EdU Staining and Migration Assays 

Human colon cancer cells were grown on coverslips incubated with SFM, oleic acid, or conditioned media from ESC alone or obese CRC-MPS and in the presence of BrdU (10 µM, Sigma, St. Louis, MO, USA). We visualized and quantified as we described before [7]. A parallel experiment utilizing EdU to visualize the proliferation of cells was performed according to the manufacturer’s instructions (C10639, Invitrogen, Waltham, MA, USA). Colon cancer cell migration confluency disruption and transwell assays were performed as described before [7]. 

### 2.6. RNA Isolation and cDNA Synthesis

RNA isolation (miRNeasy kit, Qiagen, Germany) and cDNA synthesis (SuperMix synthesis system, Thermo Fisher, Waltham, MA, USA) were performed according to the manufacturers’ protocols, as we described before [8]. 

### 2.7. Quantitative PCR 

cDNA generated from human colon cancer cells and colonospheres was utilized for qPCR as previously described [8]. Independent experiments were performed 3 times by different investigators. The primers for amplification of human cDNA were: for hSNAI1 (F: 5′-GGTTCTTCTGCGCTACTGCT-3′, R: 5′-TGCTGGAAGGTAAACTCTGGATT-3′), hTRIB2 (F: 5′-AGCTGGTGTGCAAGGTGTT-3′, R: 5′-GAGCAGACAGGCAAAAGCAC-3′), hCENPE (F: 5′-AGCCTGCAAGAAACCAAAGC-3′, R: 5′-TCTGTCGGTCCTGCTTTTTCT-3′), hBCCIP (F: 5′-ATGTACCAGCAGCTTCAGAAAGA-3′, R: 5′-AGTAGCACTTCCCACATGGC-3′), hNNMT (F: 5′-TGATTGACATCGGCTCTGGC-3′, R: 5′-TCTGGACCCTTGACTCTGTTC-3′), and hOAS1 (F: 5′-CTCCTGGATTCTGCTGACCC-3′, R: 5′-GTGCAGGTCCAGTCCTCTTC-3′). The relative levels of mRNA were determined by the comparative Ct method using actin and GAPDH as housekeeping controls as previously described [7,8].

### 2.8. RNA Sequencing and Differential Expression 

RNA sequencing (RNAseq) of experimental colon cancer cells was accomplished as described previously [7,8]. Transcriptomic data and differentially expressed genes (DEGs) were analyzed by using Ingenuity Pathway Analysis (IPA, Qiagen, Germany). Principal component analysis (PCA) and unsupervised hierarchal clustering were performed as we described before [7,8].

### 2.9. Statistical Analysis

All experiments were repeated independently by the same or different researchers, and data are represented as mean ± S.D. for a series of experiments. Investigators were blinded during experimentation. Student’s unpaired t-test or one-way analysis of variance (ANOVA) and a Student–Newman–Keuls post-test were each calculated as we described before [7,8]. Confidence intervals (95%) were calculated for human samples as well.

## 3. Results and Discussion 

The role of EPAT in homeostasis and pathobiology of the colon is understudied, primarily due to the lack of models and challenges in obtaining biopsies. We established a novel microphysiological system (MPS) utilizing human EPAT and colonic cells sandwiched between tissue-engineered sheets of EPAT-derived stromal cells (ESC) (Figure 1A). It is important to highlight that in this EPAT-dependent colorectal cancer model (CRC-MPS), which is stable in culture for a week, wholesale EPAT tissue is used, not isolated nor differentiated adipocytes. As human white adipose tissue sandwiched between sheets of stromal cells maintained physiologic tissue characteristics [16], the CRC-MPS provides a reliable ex vivo model to study EPAT-mediated processes in colonic cells. EPAT was obtained from obese (Body Mass Index (BMI) > 29.9) and lean (BMI < 24.9) patients undergoing surgery unrelated to colonic inflammation or cancer. Initially, in this CRC-MPS, we utilized HT29-GFP cells, as a reporter and stable colon cancer cell line, to establish a reliable model. In the CRC-MPS model, we found significantly increased growth of HT29-GFP cells when co-cultured with obese EPAT compared to the control, represented by HT29-GFP sandwiched between the ESC sheets alone (Figure 1B). When co-cultured with EPAT obtained from lean individuals, HT29-GFP growth remained unaffected compared to ESC alone (Figure 1B). Further, confidence intervals (CI, 95%) and a box and whisker plot demonstrated a 4.895-fold increase in colon cancer HT29-GFP cell growth by obese EPAT (relative to lean) normalized to ESC-only negative controls (CI, obese: (3.79, 5.21), lean: (0.68, 1.16)) (Figure 1C). Moreover, this finding was supported using conditioned media (CM) from obese CRC-MPS and two colon cancer cell lines (HT29 and HCT116). Specifically, this CM significantly stimulated BrdU or EdU incorporation into newly synthesized DNA in both HT29 and HCT116 cells relative to CM from ESC alone (Figure 2A,B). Additionally, we noticed in the CRC-MPS model that HT29-GFP cells were closely associated with obese—relative to lean—EPAT (Figure 1A). We assessed this migratory signal in CM from the obese EPAT to colon cancer cells using HT29 and HCT116 cells in confluency disruption and transwell migration assays. After the confluency disruption of HT29 cells, CM presence led to a smaller distance between their migratory fronts (Figure 2C). Further, it increased HCT116 colon cancer cell migration from the upper chamber to the lower chamber of transwells (Figure 2D). These findings revealed the important role of EPAT in augmenting colon cancer cell growth and migration with obesity (directly and indirectly). Utilization of two colon cancer cell lines may limit interpretation; hence, different colonic cells will be considered in further expansion of the CRC-MPS. Further, metabolic remodeling has a profound effect on the transcriptomes of colonic cells compared to underlying mutations [21]; thus, we speculate that parental gene mutations will have a secondary effect on metabolic remodeling, requiring further exploration of critical mechanisms driving tumorigenesis. 

Next, we sought to determine how EPAT mediates growth and migratory behavior by systematically surveying the gene expression in colon cancer cells (RNA-seq). We identified 867 differentially expressed genes (DEGs) altered in HT29 cells by CM from obese CRC-MPS (>|1.5|-fold change, FDR < 0.05 and meeting stringent differential expression and statistical thresholds of log_2_ fold-change > |1.5| and an adjusted *p*-value < 0.001). These DEGs, representing a transcriptional signature mediated by human obese EPAT in colon cancer cells (hOEC_867_), are linked to cancer, gastrointestinal diseases, metabolic processes, and growth (Figure 3A). Moreover, the pathways representing hOEC_867_ were compared with pathways representing transcriptomes of the colon of high-fat-diet (HFD) obese mice and colonic tumors of HFD obese mice with AOM/DSS-induced tumors. In this mouse model, we demonstrated that HFD obese mice had increased colonic tumor burden and pathways associated with metabolic and tumorigenic remodeling [8]. We found that pathways representing hOEC_867_ were similar to the pathways in the colon of HFD obese mice in colonic tissue (Figure 3B, IPA) and tumors (and when compared to the colon of mice fed with a regular diet) (Figure 3C, IPA). Further, we determined the significance of these hOEC_867_ pathways in human colonic tumorigenesis using publicly available transcriptomic data from tumor tissue samples obtained from two colon cancer patient cohorts. Pathways representing hOEC_867_ were similar to those characterizing human colon cancer (GSE4183, GSE141174) (Figure 3D, IPA). Next, we determined the significance of hOEC_867_ in human colonic tumorigenesis using publicly available transcriptomes from a large patient population (TCGA). Principal component analysis (PCA) and unsupervised hierarchal clustering showed that hOEC_867_ separated the transcriptomes of human colon cancer samples from normal with a high degree of significance (*p =* 9.6 × 10^−11^) (TCGA, Figure 3E,F). These findings demonstrated the importance of EPAT-mediated gene expression in colonic cells in obesity-augmented metabolic and tumorigenic remodeling.

Finally, we validated selected obese EPAT-mediated DEGs for expression in HT29 cells treated with CM from obese CRC-MPS vs. ESC (Figure 4A). Transcriptional levels of these genes were similarly altered in the human colon cancer samples relative to normal (TCGA, Figure 4A). These genes regulate diverse cellular functions, and limited data implicated them in cancer. Specifically, SNAI1 is linked to the epithelial-to-mesenchymal transition (EMT), while TRIB2 is associated with colon cancer stem cells [22,23]. Further, several of these genes regulate the cell cycle and migration, such as CENPE, a kinesin-like motor protein required for stable spindle microtubule capture, and BCCIP, which modulates CDK2 kinase activity [24,25]. OAS1 was recently detected in pancreatic and breast cancer [26,27]. Moreover, NNMT regulates methylation and has been linked to gastric and colon cancer [28]. Next, we analyzed the distribution of these genes’ expression according to the BMI status of colon cancer patients, utilizing cancer stage subsets based on clinical parameters associated with tumor dissemination [29]. In boxplots corresponding to significant and close-to-significant differential gene expression, we found increased levels of TRIB2 and CENP2 corresponding with obesity (BMI > 30) and lymph node metastasis (Figure 4B). Similarly, increased NNMT levels correspond to obesity in late stages of solid colonic and metastatic tumors (lymph nodes, perineural invasion, and distant metastasis) (Figure 4B). These findings demonstrated that in obese individuals, EPAT mediates the expression of genes in colonic cells associated with growth, cancer stem cells, and epigenetic changes (methylation). Further, increased levels of several of these genes regulating cancer stem cells, migration, and methylation, corresponding to obesity, are linked to metastasis, which establishes a platform to determine novel biomarkers representing obese EPAT-mediated colorectal tumorigenesis. 

Here, we demonstrated a tumor-promoting role of human EPAT in obesity-facilitated colorectal cancer utilizing our novel model. Emerging findings demonstrated that EPAT in obese individuals differs from mesenteric, omental, and subcutaneous fats in methylome, transcriptome, and proteome [9]. Further, EPAT has uniquely altered pathways in obese individuals with insulin resistance compared to insulin sensitivity [9], suggesting its systemic metabolic role. This adipose tissue is understudied due to lack of a mouse model and the difficulty of obtaining human EPAT tissue biopsies. Our novel ex vivo model will allow us to further elucidate mechanisms of EPAT-mediated processes in the colon. Utilization of wholesale EPAT tissue gives us an advantage for future understanding of the EPAT cell landscape (adipose, non-adipose, and stromal cells) and their released mediators in processes involving the colon. As the initial utilization of colon cancer cells in this model can be extended to human organoids, it is important to keep in mind that the complexity of colonic tissue is underrepresented, primarily as other non-colonic cells are not considered.

Moreover, our findings revealed that EPAT may indirectly and directly augment tumorigenic processes in colonic cells, in part, by impacting the tumor microenvironment. Onogi and Ussar suggested a possible role of EPAT in the regulation of host–microbe interaction [13]. In the small intestine, microbiota translocate to mesenteric adipose tissue, which promotes M2 macrophage activation [30]. Therefore, it is plausible that in the colon, interactions between EPAT and microbiota may be one of the mechanisms driving tumorigenesis in obesity. Moreover, microbiota and obesity facilitate inflammatory signaling in the colon, which may be essential for EPAT function, as studies have shown that inflammation is required for physiological adipose tissue remodeling [31]. Additionally, released mediators from EPAT (adipocytes and non-adipocytes), such as TNF and leptin, as well as metabolites, such as fatty acids [8,9,32,33], may further drive tumorigenesis in the colon. Together, EPAT and microbiota axes could create a microenvironment that is further augmented with obesity in promoting tumorigenesis in the colon. Along with these complex changes in the colonic environment, EPAT may directly affect colonic cells, such as their gene expression and epigenetic changes. We found that methylation via NNMT is augmented in colonic cells by EPAT. Further, EPAT-mediated growth of colon cancer cells may be, in part, due to CENPE- and BCCIP-dependent cell cycle dysregulation. Our findings further revealed that EPAT may affect cancer stem cells, in part, via TRIB2, which can lead to resistance to therapy and tumor recurrence in colon cancer patients. Moreover, EPAT may accelerate the metastatic characteristics of colon cancer cells by promoting EMT and the spread of cancer cells. It is possible that EPAT drives EMT through increased expression of SNAI1, which may involve loss of FOXO3 in colonic cells mediated by obesity. In the colon, loss of FOXO3 is one of the mechanisms by which obesity mediates metabolic reprogramming linked to tumorigenic processes [6,34,35]. In renal cell carcinoma, loss of FOXO3 facilitates EMT by increasing SNAI1 [36]. Moreover, increased expression of genes regulating cancer stem cells, migration, and methylation (TRIB2, CENP2, NNMT) corresponded to metastatic tumors in obese colon cancer patients. Therefore, it is tempting to speculate that EPAT may aid escaped metastatic colon cancer through lymph nodes to distant organs. These findings highlight the critical role of EPAT in obesity-facilitated colonic tumorigenesis, thus establishing a platform to identify novel markers and to develop effective treatment options.

## 4. Conclusions

The obesity epidemic, affecting half a billion individuals worldwide, is associated with increased CRC risk, progression, recurrence, resistance to therapy, and mortality [1,3]. Further, CRC incidence has been on the rise globally among young adults, mainly due to this epidemic [37,38]. This poses an urgent, unmet demand to understand the mechanisms driving obesity-mediated tumorigenesis in the colon. Here, we demonstrated that fat outpouchings attached to the colon, known as EPAT, have a tumor-promoting role in obesity-facilitated colonic tumorigenesis, especially in metastatic processes. These findings provide conceptual advances in our understanding of how obesity facilitates CRC. Establishing this platform will further drive the exploration of critical mechanisms of EPAT-mediated processes in colonic cells and identify novel biomarkers needed for the development of effective treatment options for CRC. 

## Figures and Tables

**Figure 1 cancers-15-00977-f001:**
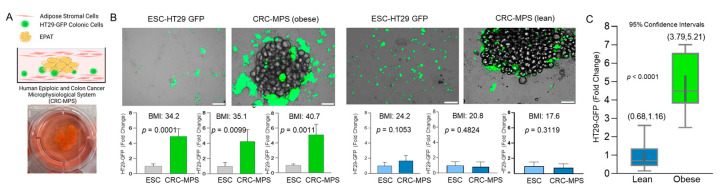
(**A**) Cross-sectional model of Epiploic (EPAT) Colorectal Cancer Microphysiological System (CRC-MPS). (**B**) Representative images of CRC-MPS with HT29-GFP cells and EPAT obtained from obese (BMI 34.2) or lean (BMI 24.2) individuals vs. HT29-GFP grown with EPAT-derived stromal cells (ESC) only (48 h). Graphs represent the intensity of GFP pixels from CRC-MPS images, three independent wells for each EPAT, obtained from three obese or three lean individuals relative to ESC (Scale bar = 100 µm, BMI *=* Body Mass Index). (**C**) All individual values from CRC-MPS were normalized to matched ESC-only GFP signal, then normalized to lean average. Box and whisker plot represents the median, 25% and 75% quartiles, and highest and lowest values of each group. Thick dark grey bar within the box and whisker plot reports confidence interval (CI, 95%) with range in brackets above each box.

**Figure 2 cancers-15-00977-f002:**
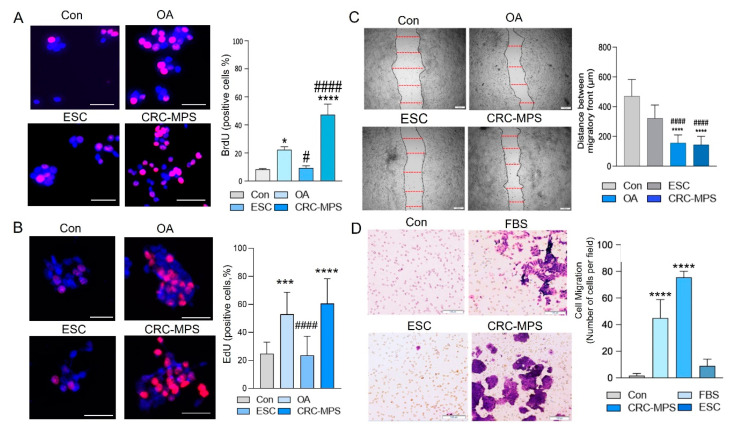
(**A**) Representative images and quantification of BrdU incorporation during DNA synthesis in HT29 cells treated with conditioned media (CM) from CRC-MPS (obese) vs. CM from ESC. Oleic acid (OA) treatment represents positive control and serum-free media (Con) represents negative control (24 h, *n =* 3, **** *p* < 0.0001, * *p* < 0.05 vs. Con; #### *p* < 0.0001, # *p* < 0.05 vs. OA, scale bar = 50 μm). (**B**) Representative images and quantification of EdU incorporation during DNA synthesis in HCT116 cells treated with CM from CRC-MPS (obese) vs. CM from ESC. Oleic acid (OA) treatment represents positive control and serum-free media (Con) represents negative control (24 h, *n =* 3 wells in 2 independent experiments, **** *p* < 0.0001, *** *p* < 0.001 vs. Con; #### *p* < 0.0001 vs. OA, scale bar = 50 μm). (**C**) Representative images and quantification of confluency disruption of HT29 monolayers via scratch assay (24 h, *n =* 3, **** *p* < 0.0001 vs. Con; #### *p* < 0.0001 vs. ESC, scale bar 200 = μm). (**D**) Transwell migration assay of HCT116 cells, with CM from CRC-MPS (obese), ESC, fetal bovine serum (FBS) as a positive control, and serum-free media as a negative control (Con) (24 h, *n =* 3, **** *p* < 0.0001 vs. Con, scale bar 100 = μm).

**Figure 3 cancers-15-00977-f003:**
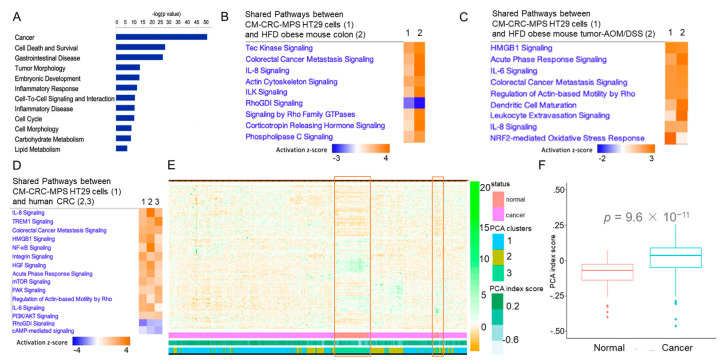
(**A**) Top diseases representing differentially expressed genes (DEGs) in colon cancer HT29 cells mediated by CM from CRC-MPS from obese patients relative to control (*p* < 0.05, IPA). (**B**,**C**) Top canonical pathways representing these DEGs compared to DEGs from colon and colonic tumors (AOM/DSS) of HFD obese mice (*n =* 3 for each group, FC > |1.5|, FDR < 0.05, IPA). In (**B**), Column 1 contains pathways representing DEGs in HT29 cells treated with CM from CRC-MPS (vs. ESC), and Column 2 contains pathways in HFD obese mouse colon (vs. RD colon). In (**C**) Column 1 contains pathways representing DEGs in HT29 cells treated with CM from CRC-MPS (vs. ESC), and Column 2 contains pathways in HFD obese mouse tumors (AOM/DSS model vs. RD normal colon). (**D**) Shared pathways representing DEGs from HT29 cells treated with CM from CRC-MPS (Column 1) and human colon cancer relative to normal colon (Column 2, GSE4183, *n =* 23; Column 3, GSE141174, *n =* 198; fold change > |1.5|, FDR < 0.05, IPA). (**E**,**F**) Unsupervised hierarchical clustering and heatmap showing hOEC_867_ signature separating transcriptome of colon cancer samples from normal. Orange boxes highlight normal samples. Principal Component Analysis (PCA) of hOEC_867_ signature and significance of PCA index score (TCGA), normal colon (*n =* 41), and tumors (*n =* 457).

**Figure 4 cancers-15-00977-f004:**
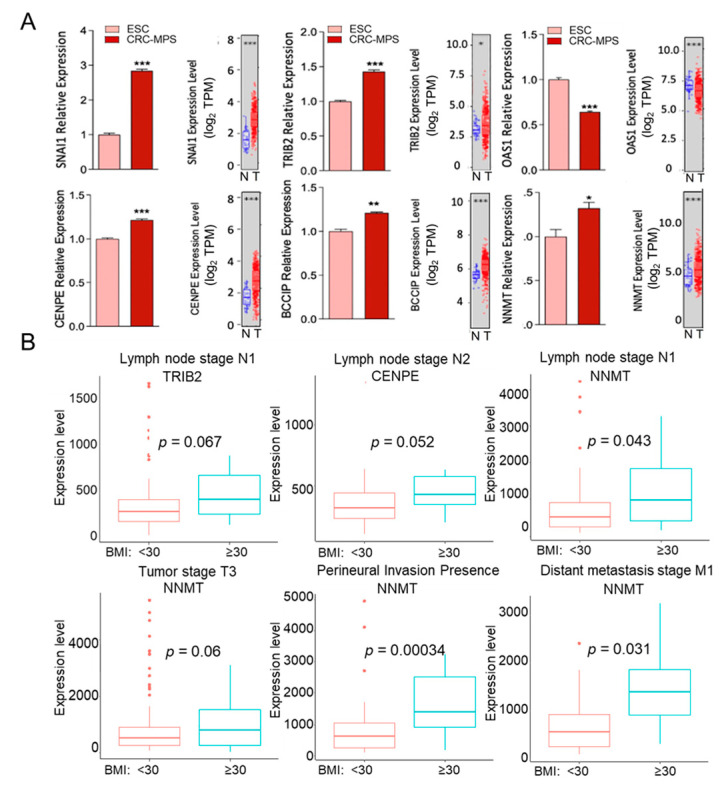
(**A**) Validation of DEGs in HT29 cells treated with CM from CRC-MPS (obese) and ESC (24 h) by qPCR. The following transcripts were assessed: SNAI1, TRIB2, OAS1, CENPE, BCCIP, and NNMT (qPCR, *n =* 3, * *p* < 0.05, ** *p* < 0.01, *** *p* < 0.001). Similar significant alterations of these transcripts were found in colon cancer patient tissue (grey box plots) (TCGA: normal colon (*n =* 41) and tumors (*n =* 457)). (**B**) Transcriptomic data of samples from colon cancer patients that included both weight and height information (TCGA) were further classified by BMI (obese: BMI > 30, *n =* 84, and non-obese: BMI < 30, *n =* 198). Selected genes were analyzed for expression levels (log_2_ TPM) in various clinical cancer stage subsets associated with tumor dissemination (Box plots).

## Data Availability

The data presented in this study are available on request from the corresponding author. RNA sequencing of experimental colon cancer cells will be submitted to NCBI’s Archive to be publicly available.

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
