# Peer review of "Epiploic Adipose Tissue (EPAT) in Obese Individuals Promotes Colonic Tumorigenesis: A Novel Model for EPAT-Dependent Colorectal Cancer Progression"

_cancers, 2023, doi:10.3390/cancers15030977_

Round 1

Reviewer 1 Report

This is an intriguing study addressing rising obesity, epiploic fat (EPAT), and CRC phenotypes. The investigators use a novel organoid system in which they co-culture EPAT from obese or non-obese patients with CRC cells and define the phenotypic consequences. They discover the HT29 cells migrate towards, proliferate and increase their migration and invasion (using scratch and trans well assays) when cultured with EPAT obtained from obese individuals. Conditioned media resulted in transcriptomic changes in metabolic and known pro-tumorigenic pathways. Validation on select targets was performed and associations were made with TCGA CRC transcriptomic data. This is a well written report describing a carefully executed serious of experiments studying the important topic of how obesity and epiploic fat contributes to CRC. I do have a few critiques that should be addressed.

Major Critiques:

  1. Could the authors provide more background on the relationship between BMI and EPAT? This is important because they are using BMI as a marker to stratify their study into the two groups. 
  2. Why were the HT29 cells selected for these experiments. It would be important to determine if these observations were cell line dependent so conducting a subset of these experiments with additional cells lines would help determine the generalizability of these findings.
  3. It is exciting to see that some of the targets induced by obese EPAT CM were also increased in TCGA data. Can the PIs stratify these results by BMI. It would further support their hypothesis if the differential expression was present in the high BMI patients.

Minor Critiques:

  1. Suggest the authors thoroughly proof the manuscript for grammatical and typographical errors. 

Reviewer 2 Report

This is an interesting study that evaluates the “Epiploic adipose tissue (EPAT) in obese individuals promotes colonic tumorigenesis: a novel model for EPAT dependent colorectal cancer progression”. However, the manuscript can be improved with some points below that the authors should consider.

1.  Due to EPAT has a tumor-promoting role in obesity facilitated colonic tumorigenesis.  What are the possible EPAT adipocyte-derived factors in microphysiological system?

2.  The limitation of CRC-MPS ex vivo model should be considered and mentioned in discussion.

Reviewer 3 Report

The study demonstrates that epiploic adipose tissue (EPAT) contributes in promoting colonic tumorigenesis. Microphysiological system (MPS) has been developed in the study.

1. Introduction may be revised to highlight the EPAT more in terms of MPS model development.

2. Method needs to be revised to indicate that the data on mice were transcriptomic data in 2.2. Mice. The sub-title of the section may be changed. The information on the data whether it is public or non-public may be added.

3. Figure 3 may be revised to make the differences between B and C clearer.

4. Discussion and Conclusion needs to be revised to highlight the role of EPAT in cancer promotion and possible involvement of epithelial mesenchymal transition with additional references.  

Round 2

Reviewer 3 Report

The authors revised the manuscript according to the reviewers' comments. It has been greatly improved.